# Lung Mycobiota $\alpha$-Diversity Is Linked to Severity in Critically Ill Patients with Acute Exacerbation of Chronic Obstructive Pulmonary Disease

Raphaël Enaud,[a,b] Pierre Sioniac,[c] Sebastien Imbert,[a,d] Pierre-Laurent Janvier,[c] Adrian Camino,[a] Hoang-Nam Bui,[c] Odile Pillet,[c] Arthur Orieux,[c] Alexandre Boyer,[a,c] Patrick Berger,[a] Didier Gruson,[a,c] Laurence Delhaes,[a,d] Renaud Prevel[a,c]

[a]Université Bordeaux, Center de Recherche Cardio-Thoracique de Bordeaux, INSERM UMR 1045, Bordeaux, France
[b]Centre Hospitalier Universitaire Bordeaux, Le Centre de Ressources et de Compétences de la Mucoviscidose Pédiatrique, CIC 1401, Bordeaux, France
[c]Centre Hospitalier Universitaire Bordeaux, Medical Intensive Care Unit, Bordeaux, France
[d]Centre Hospitalier Universitaire Bordeaux, Mycology-Parasitology Department, CIC 1401, Bordeaux, France

**ABSTRACT** Chronic obstructive pulmonary disease (COPD) affects more than 200 million people worldwide. The chronic course of COPD is frequently worsened by acute exacerbations (AECOPD). Mortality in patients hospitalized for severe AECOPD remains dramatically high, and the underlying mechanisms are poorly understood. Lung microbiota is associated with COPD outcomes in nonsevere AECOPD, but no study specifically investigated severe AECOPD patients. The aim of this study is thus to compare lung microbiota composition between severe AECOPD survivors and nonsurvivors. Induced sputum or endotracheal aspirate was collected at admission from every consecutive severe AECOPD patient. After DNA extraction, the V3-V4 and ITS2 regions were amplified by PCR. Deep-sequencing was performed on a MiSeq sequencer (Illumina); the data were analyzed using DADA2 pipeline. Among 47 patients admitted for severe AECOPD, 25 (53%) with samples of sufficient quality were included: 21 of 25 (84%) survivors and 4 of 25 (16%) nonsurvivors. AECOPD nonsurvivors had lower $\alpha$-diversities indices than survivors for lung mycobiota but not for lung bacteriobiota. Similar results were demonstrated comparing patients receiving invasive mechanical ventilation ($n = 13$ [52%]) with those receiving only noninvasive ventilation ($n = 12$ [48%]). Previous systemic antimicrobial therapy and long-term inhaled corticosteroid therapy could alter the lung microbiota composition in severe AECOPD patients. In acidemic AECOPD, lower lung mycobiota $\alpha$-diversity is linked to the severity of the exacerbation, assessed by mortality and the requirement for invasive mechanical ventilation, whereas lung bacteriobiota $\alpha$-diversity is not. This study encourages a multicenter cohort study investigating the role of lung microbiota, especially fungal kingdom, in severe AECOPD.

**IMPORTANCE** In AECOPD with acidemia, more severe patients—*i.e.*, nonsurvivors and patients requiring invasive mechanical ventilation—have lower lung mycobiota $\alpha$-diversity than survivors and patients receiving only noninvasive ventilation, respectively. This study encourages a large multicenter cohort study investigating the role of lung microbiota in severe AECOPD and urges investigation of the role of the fungal kingdom in severe AECOPD.

**KEYWORDS** chronic obstructive pulmonary disease, severe exacerbation, microbiota, mycobiota, intensive care unit, human microbiota

Address correspondence to Renaud Prevel, renaud.prevel@hotmail.fr, or Raphaël Enaud, raphael.enaud@chu-bordeaux.fr.

The authors declare no conflict of interest.

Chronic obstructive pulmonary disease (COPD) is a highly prevalent disease affecting more than 200 million people worldwide (1) and may become the fourth leading cause of death worldwide in 2030 (2). Its main risk factors are tobacco smoking and

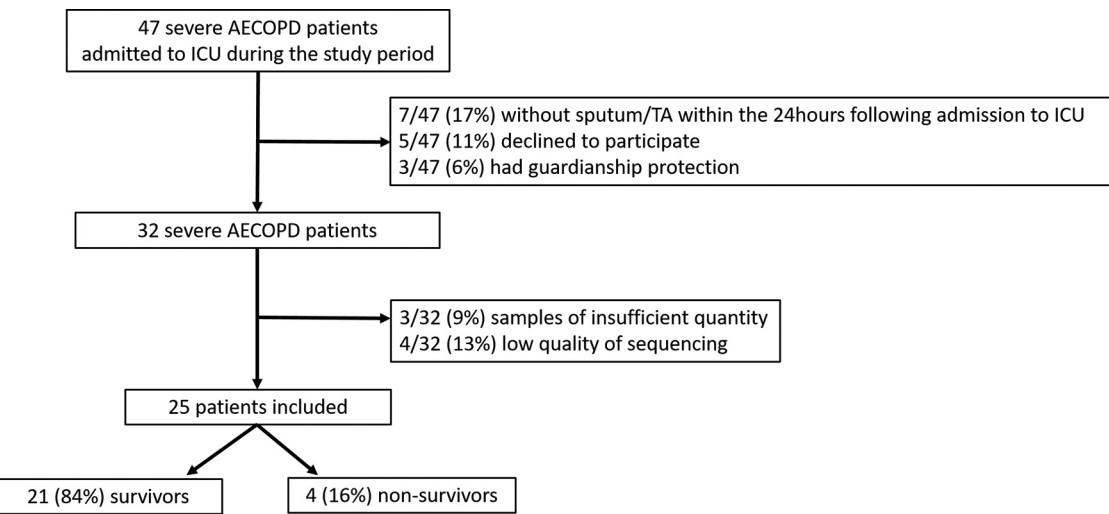

**FIG 1** Flow chart. AECOPD, acute exacerbation of chronic obstructive pulmonary disease; ICU, intensive care unit; TA, tracheal aspirate.

air pollution (1). The course of this chronic disease is frequently worsened by acute exacerbations (AECOPD) in nearly 80% of COPD patients (3). AECOPD are often related to viral or bacterial infections (4), and despite the description of various sputum cellular profiles (5), the mechanisms underlying AECOPD remain mostly unknown. The frequency of AECOPD is mainly dependent on the occurrence of previous infections, and they dramatically increase the morbidity and mortality of these patients (3, 6). In fact, patients with frequent AECOPD exhibit a faster decline in lung function and a higher mortality rate (6), especially in cases of hospitalization (7), than those without. Hospitalized AECOPD patients have a 45% mortality rate within the 4 years following hospitalization (8). More severe AECOPD causes hypercapnia and acidemia requiring admission to an intensive care unit (ICU). During the last decade, the lung microbiota has been linked to numerous acute and chronic respiratory diseases (9), including COPD regarding both chronic severity (10–12) and occurrence of AECOPD (12, 13). Nevertheless, no study so far has focused on the link between lung microbiota composition and survival in severe AECOPD patients, i.e., causing acidemia requiring admission to the ICU. Moreover, most of these studies focused on the bacterial kingdom of the lung microbiota, and only one study investigated the fungal kingdom (14), whereas fungi are known to be involved in many respiratory diseases (9). The aim of this study is thus to compare the lung bacterial and fungal microbiota (bacteriobiota and mycobiota) composition between severe AECOPD survivors and nonsurvivors.

## RESULTS

**Flow chart and patients' characteristics.** Among 47 patients admitted for severe AECOPD, 25 (53%) were included in the final analysis: 21 of 25 (84%) of ICU survivors and 4 of 25 (16%) of ICU nonsurvivors (Fig. 1). Patient characteristics are presented in Table 1. Briefly, survivors and nonsurvivors had similar characteristics at admission, but nonsurvivors received more frequent long-term triplet inhaled therapy than survivors ($P = 0.05$). The proportion of patients who received long-term inhaled corticosteroid therapy and systemic antibiotics was similar between survivors and nonsurvivors ($P = 0.12$ and $P = 1.00$, respectively). Survivors had higher arterial carbon dioxide arterial partial pressure ($PaCO_2$) and bicarbonate levels than nonsurvivors ($P = 0.04$ and $P = 0.02$, respectively) resulting in similar arterial pH ($P = 0.94$). All patients were directly admitted to the ICU after admission to hospital.

**Past systemic antimicrobial therapy and long-term inhaled corticosteroids could affect lung microbiota composition in severe AECOPD patients.** As systemic corticosteroids or antimicrobial therapies within the previous month or long-term

**TABLE 1** Patient characteristics and comparison between severe AECOPD survivors and nonsurvivors[a]

| Characteristic | Total (n = 25) | Nonsurvivors (n = 4) | Survivors (n = 21) | P value |
|---|---|---|---|---|
| **Patients' characteristics at admission to ICU** | | | | |
| Age | 66 [62 to 72] | 65 [58 to 70] | 66 [62 to 72] | 0.60 |
| Sex (male) | 14 (56%) | 3 (75%) | 11 (52%) | 0.60 |
| Gold score (n = 20) | | | | 0.65 |
| 3 | 10 (50%) | 2/3 (67%) | 8/17 (47%) | |
| 4 | 10 (50%) | 1/3 (33%) | 9/17 (53%) | |
| Home oxygen therapy | 13 (52%) | 1 (25%) | 12 (57%) | 0.32 |
| Past tobacco smoking | 25 (100%) | 4 (100%) | 21 (100%) | 1.00 |
| Current tobacco smoking | 7 (28%) | 0 (00%) | 7 (33%) | 0.29 |
| Previous admission to ICU for AECOPD | 11 (44%) | 1 (25%) | 10 (48%) | 0.60 |
| Chronic heart failure | 5 (20%) | 1 (25%) | 4 (19%) | 1.00 |
| Chronic coronary disease | 5 (20%) | 1 (25%) | 4 (19%) | 1.00 |
| Immunosuppression | 3 (12%) | 0 (00%) | 3 (14%) | 1.00 |
| COPD medications | | | | |
| Inhaled treatments | | | | |
| B2 agonists | 17 (68%) | 3 (75%) | 14 (66%) | 1.00 |
| Anticholinergics | 16 (64%) | 3 (75%) | 13 (62%) | 1.00 |
| Corticosteroids | 9 (36%) | 3 (75%) | 6 (29%) | 0.12 |
| Dual therapy | 9 (36%) | 0 (00%) | 9 (43%) | 0.26 |
| Triplet therapy | 7 (28%) | 3 (75%) | 4 (19%) | 0.05 |
| Systemic corticosteroids in the previous mo | 5 (20%) | 1 (25%) | 4 (19%) | 1.00 |
| Systemic antibiotics in the previous mo | 6 (24%) | 1 (25%) | 5 (24%) | 1.00 |
| **AECOPD presentation** | | | | |
| SAPS II | 50 [37 to 67] | 57 [55 to 69] | 47 [35 to 54] | 0.06 |
| Heart rate (/min) | 94 [81 to 113] | 120 [85 to 155] | 94 [78 to 107] | 0.14 |
| Systolic blood pressure (mm Hg) | 129 [117 to 152] | 141 [124 to 189] | 129 [115 to 151] | 0.22 |
| Diastolic blood pressure (mm Hg) | 70 [64 to 80] | 75 [62 to 103] | 70 [59 to 78] | 0.46 |
| Respiratory rate | 28 [20 to 33] | 28 [26 to 33] | 28 [20 to 33] | 0.82 |
| Accessory muscles activation | 17 (68%) | 2 (50%) | 15 (71%) | 0.57 |
| pH | 7.25 [7.15 to 7.31] | 7.27 [7.15 to 7.34] | 7.25 [7.15 to 7.31] | 0.94 |
| $PaCO_2$ (kPa) | 9.9 [8.2 to 13.1] | 7.8 [7.62 to 8.8] | 10.2 [8.7 to 14.0] | 0.04 |
| $PaO_2$ (kPa) | 9.5 [7.9 to 12.5] | 9.2 [7.2 to 14.8] | 9.5 [7.9 to 12.5] | 1.00 |
| Bicarbonates (mmol/L) | 32.4 [28.6 to 35] | 26.2 [22.8 to 31.2] | 34 [30.8 to 36.7] | 0.02 |
| **Treatment** | | | | |
| Noninvasive ventilation at any time | 24 (96%) | 3 (75%) | 21 (100%) | 0.16 |
| Invasive mechanical ventilation | 13 (52%) | 4 (100%) | 9 (43%) | 0.09 |
| Tidal volume (mL/kg of ideal body wt) | 6.8 [6.2 to 7.8] | 7.2 [6.6 to 9.2] | 6.5 [5.9 to 7.9] | 0.40 |
| Ventilation rate (/min) | 24 [19 to 26] | 25 [23 to 27] | 24 [19 to 27] | 0.55 |
| Inspired fraction of $O_2$ (%) | 32 [25 to 40] | 40 [30 to 55] | 30 [25 to 40] | 0.35 |
| Positive end expiratory pressure ($cmH_2O$) | 6 [5 to 6] | 7 [5 to 11] | 5 [5 to 6] | 0.15 |
| Neuromuscular blockade | 4/13 (31%) | 2/4 (50%) | 2/9 (22%) | 0.22 |
| **Outcomes** | | | | |
| Day 28 mortality | 4 (16%) | 4 (100%) | 0 (0%) | |
| Length of noninvasive ventilation (days) | 4 [3 to 6] | 10 [4 to 11] | 4 [3 to 6] | 0.07 |
| Length of invasive ventilation (days) | 5 [4 to 13] | 43 [12 to 52] | 5 [4 to 6] | 0.05 |
| ICU length of stay (days) | 8 [4 to 11] | 51 [13 to 62] | 7 [5 to 9] | 0.06 |
| Hospital length of stay (days) | 14 [8 to 26] | 72 [18 to 95] | 11 [8 to 20] | 0.18 |

[a]The results are presented as proportions for categorical variables and median [interquartile range] for continuous variables The P values are for comparison between survivors and nonsurvivors. The threshold for statistical significance was P = 0.05. AECOPD, acute exacerbation of chronic obstructive pulmonary disease; COPD, chronic obstructive pulmonary disease; ICU, intensive care unit; $PaCO_2$, carbon dioxide arterial partial pressure; $PaO_2$, dioxygen arterial partial pressure; SAPS II, simplified acute physiology score II.

inhaled corticosteroid therapy could alter the lung microbiota composition, we compared patients who did or did not receive those treatments. Despite not reaching statistical significance, systemic antimicrobial therapy (amoxicillin alone or amoxicillin + clavulanic acid) within the previous month could have induced dissimilarity in lung mycobiota between patients who did or did not receive it (permutational multivariate analysis of variance [PERMANOVA], P = 0.06; Fig. S4) but not in lung bacteriobiota (PERMANOVA, P = 0.27; Fig. S5). Long-term inhaled corticosteroid therapy does not

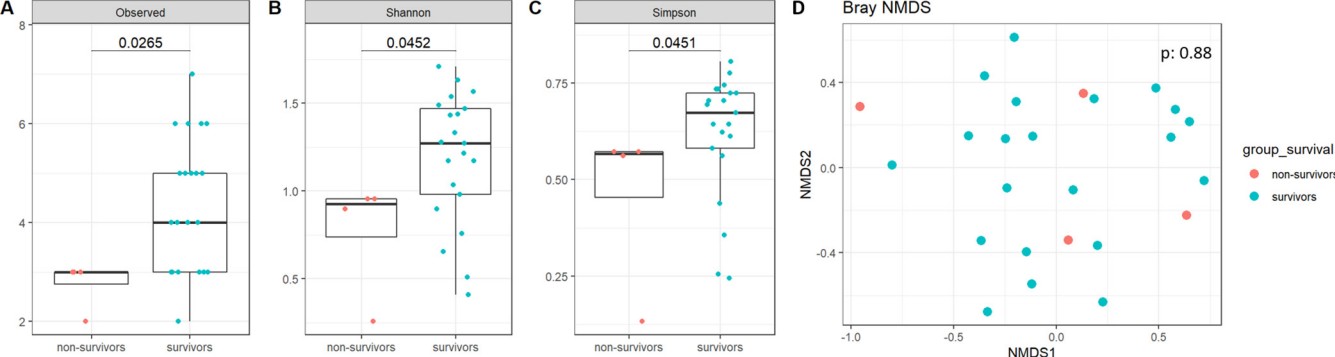

**FIG 2** Comparison of lung mycobiota between survivors and nonsurvivors in severe AECOPD patients. (A) Boxplot of estimated α-diversity by Shannon index. (B) Boxplot of estimated α-diversity by Simpson index. (C) Boxplot of estimated α-diversity by evenness. (D) Nonmetric Bray-Curtis analysis of β-diversity. The threshold for statistical significance was P = 0.05.

seem to alter lung mycobiota composition (Fig. S6) but could lower lung bacteriobiota α-diversity (P = 0.13, P = 0.08, P = 0.17; Fig. S7). Systemic corticosteroid therapy within the previous month does not seem to alter lung mycobiota nor bacteriobiota composition in severe AECOPD patients (Fig. S8 and S9).

**Description of the lung mycobiota and bacteriobiota in severe AECOPD patients.** The more abundant fungal phyla were *Basidiomycota* and *Ascomycota* without any difference between survivors and nonsurvivors (Fig. S10), and the more abundant bacterial phyla were *Firmicutes*, *Proteobacteria*, *Bacteroidota*, and *Actinobacteria* without any difference between survivors and nonsurvivors (Fig. S11). In particular, the abundances of *Haemophilus* spp., *Moraxella* spp., and *Pseudomonas* spp. were not different between survivors and nonsurvivors (P = 0.45, P = 0.60, and P = 0.84, respectively).

**Severe AECOPD nonsurvivors exhibit a lower α-diversity of lung mycobiota than survivors.** The lung mycobiota α-diversity was lower in severe AECOPD nonsurvivors compared to survivors (P = 0.03, P = 0.05, and P = 0.05, respectively, for the Richness, Shannon, and Simpson indices; Fig. 2A to C), but no significant difference was observed in the lung bacteriobiota α-diversity (P = 0.71, P = 0.54, P = 0.70, respectively; Fig. 3A to C). Moreover, regarding β-diversity, both lung mycobiota and bacteriobiota were not dissimilar between severe AECOPD survivors and nonsurvivors (PERMANOVA, P = 0.88 and P = 0.91, respectively) (Fig. 2D and 3D).

**Severe AECOPD patients requiring invasive mechanical ventilation exhibit a lower α-diversity of lung mycobiota than those who required noninvasive ventilation (NIV) only.** Similar results were obtained comparing lung mycobiota α-diversity indices (P = 0.02, P = 0.02, and P = 0.03, respectively, for the Richness, Shannon and Simpson indices; Fig. 4A to C) and lung bacteriobiota α-diversity indices (P = 0.32, P =

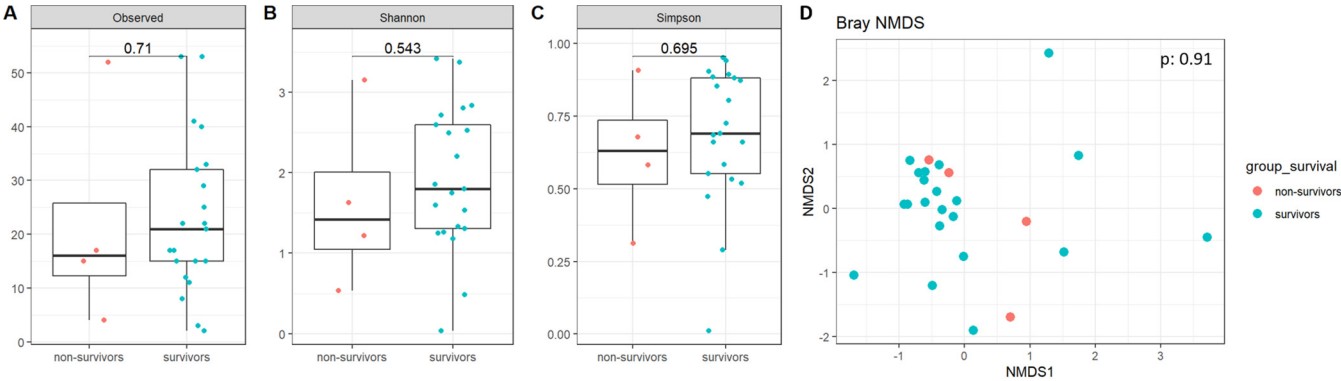

**FIG 3** Comparison of lung bacteriobiota between survivors and nonsurvivors in severe AECOPD patients. (A) Boxplot of estimated α-diversity by Shannon index. (B) Boxplot of estimated α-diversity by Simpson index. (C) Boxplot of estimated α-diversity by evenness. (D) Nonmetric Bray-Curtis analysis of β-diversity. The threshold for statistical significance was P = 0.05.

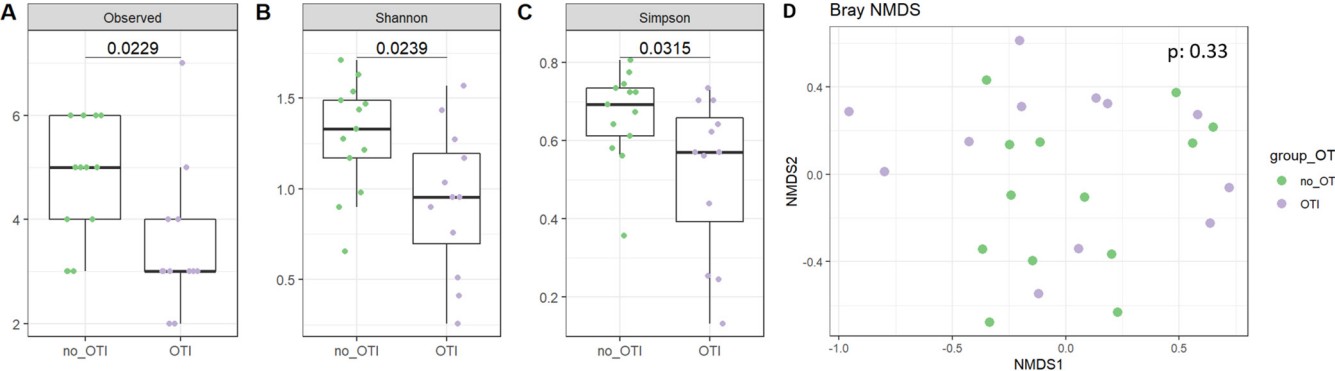

**FIG 4** Comparison of lung mycobiota between severe AECOPD patients requiring or not requiring invasive mechanical ventilation. (A) Boxplot of estimated $\alpha$-diversity by Shannon index. (B) Boxplot of estimated $\alpha$-diversity by Simpson index. (C) Boxplot of estimated $\alpha$-diversity by evenness. (D) Nonmetric Bray-Curtis analysis of $\beta$-diversity. The threshold for statistical significance was $P = 0.05$. OTI, orotracheal intubation.

0.27, and $P = 0.41$, respectively; Fig. 5A to C) between severe AECOPD patients requiring invasive mechanical ventilation or not. Consistent with the comparison between survivors and nonsurvivors, both lung bacteriobiota and mycobiota were not dissimilar between severe AECOPD patients requiring invasive mechanical ventilation or not (PERMANOVA, $P = 0.50$ and $P = 0.33$, respectively) (Fig. 4D and 5D).

## DISCUSSION

To the best of our knowledge, this study is the first to assess the link between the lung microbiota composition and survival in the specific population of critically ill patients admitted to the ICU for severe AECOPD. Notably, we identified that, in AECOPD with acidemia, more severe patients—i.e., nonsurvivors and patients requiring invasive mechanical ventilation—have lower lung mycobiota $\alpha$-diversity, but not bacteriobiota, than less severe ones—survivors and patients not requiring invasive mechanical ventilation, respectively. Systemic antimicrobial therapy and long-term inhaled corticosteroid therapy could affect the lung microbiota composition in severe AECOPD patients requiring admission to the ICU.

Thanks to the advances of next-generation sequencing, the role of both gut and lung microbiota in COPD has been investigated since the 2010s. Studies comparing lung bacteriobiota from COPD patients to healthy subjects described a change in microbial diversity with increased relative abundances of *Moraxella*, *Streptococcus*, *Veillonella*, *Eubacterium*, and *Prevotella* spp. in disease (15). This difference was not explained by the smoking history but rather by the COPD endotypes of the patients (16). Decreased lung bacteriobiota diversity with concomitant expansion of bacteria from the *Proteobacteria* phylum have been demonstrated to be associated with both increased chronic COPD

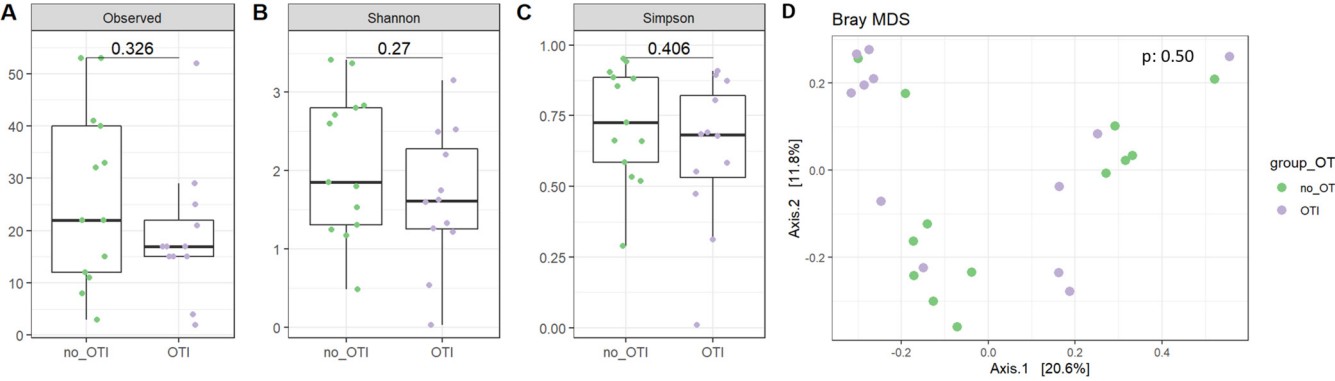

**FIG 5** Comparison of lung bacteriobiota between severe AECOPD patients requiring or not requiring invasive mechanical ventilation. (A) Boxplot of estimated $\alpha$-diversity by Shannon index. (B) Boxplot of estimated $\alpha$-diversity by Simpson index. (C) Boxplot of estimated $\alpha$-diversity by evenness. (D) Nonmetric Bray-Curtis analysis of $\beta$-diversity. The threshold for statistical significance was $P = 0.05$. OTI, orotracheal intubation.

severity and increased frequency of AECOPD (12, 17–19). More precisely, in stable COPD patients, reduced lung bacteriobiota α-diversity, associated with *Proteobacteria* (predominantly *Haemophilus* sp.) dominance, is linked to a neutrophil-associated COPD endotype and increased 4-year mortality (20). In patients hospitalized with nonsevere AECOPD, 1-year nonsurvivors also exhibit lower α-diversity and dissimilarity compared to survivors. In this study, survivors had a higher relative abundance of *Veillonella*, and nonsurvivors had a higher relative abundance of *Staphylococcus* (21). Beyond these association studies, patients with genetic mannose-binding lectin deficiency exhibited a more diverse lung microbiota and a lower risk of AECOPD, suggesting a causality link between lung microbiota and AECOPD (22). One study investigated the dynamics of lung bacteriobiota in critical AECOPD patients receiving invasive mechanical ventilation and its impact on the weaning process. An increased relative abundance of *Acinetobacter baumanii* complex species was observed in patients who failed the weaning process compared to those who succeeded. In contrast, no significant difference in α-diversity between these groups was observed (23). Nevertheless, AECOPD patients were mixed with patients with community-acquired pneumonia, thus limiting the interpretation regarding the mortality outcome specifically in severe AECOPD patients. The absence of observed difference in both lung bacteriobiota α- and β-diversities between survivors and nonsurvivors in our study could be due to the fact that we focused only on the more severe AECOPD patients who require admission to the ICU. They might all have had pre-existing highly disturbed lung bacteriobiota, which may explain the absence of observed differences.

The link between lung mycobiota and AECOPD has been poorly studied despite the fact that fungi are involved in numerous chronic respiratory diseases, bacterial and fungal kingdoms directly interact within the microbiota (9), and commensal fungi also modulate local and systemic immunity (24). A recent study demonstrated that the lung mycobiota of the subgroup of COPD patients with frequent exacerbations and higher 2-year mortality was characterized by the dominance of *Aspergillus*, *Curvularia*, and *Penicillium* (14). Consistent with these findings, we demonstrate that the more severe patients (i.e., those requiring invasive mechanical ventilation and nonsurvivors) have decreased lung mycobiota α-diversity compared to others (patients treated with NIV and survivors). Lung mycobiota dysbiosis could add up to pre-existing lung bacteriobiota alteration in this subset of very severe patients having deleterious synergistic effects. This study urges systematic assessment of lung mycobiota in future studies investigating the role of lung microbiota in COPD pathophysiology.

The main limitations of our study are its monocentric character and the limited number of nonsurvivors. Nevertheless, the fact that these findings are reproduced in a bigger population comparing patients requiring invasive mechanical ventilation or not enhances their validity.

A second possible bias was introduced by combining data from induced sputum and tracheal aspirate (TA) (25), but we did not find any significant dissimilarity between these two kinds of samples. We consistently performed induced, and not spontaneous, sputum. Moreover, if sputum provided low-quality samples, they should exhibit lower α-diversity than TA, and our analyses provided the opposite results.

A third limitation is the potential impact of previous systemic antimicrobial therapy and inhaled corticosteroids (26, 27), which can affect lung microbiota composition. As our analysis finds that long-term inhaled corticosteroid therapy lowers lung bacteriobiota α-diversity but does not affect lung mycobiota α-diversity, the validity of the observed decrease in lung mycobiota α-diversity in nonsurvivors is supposed to be robust.

As most clinical studies, our study lacks causality demonstration. To go beyond the association links provided in this study, animal models and *in vitro* studies are needed to identify the underlying mechanisms explaining this association between lung microbiota and COPD. Such pathophysiological studies, in addition to larger prospective cohort studies, will help to better decipher the precise role of lung microbiota in severe AECOPD.

Interestingly, the gut bacteriobiota has been implicated in AECOPD as a causal player (28, 29). In fact, gut bacteriobiota during AECOPD exhibits a decreased relative abundance

of bacteria belonging to the *Firmicutes* and *Actinobacteria* phyla but an increased relative abundance of bacteria belonging to the *Bacteroidetes* and *Proteobacteria* phyla in humans (30, 31). These differences are also described in patients with stable COPD condition who have lung function decline at 1-year follow-up compared to those whose lung function remained stable (32). The causality has been suggested by the fact that transfer of gut microbiota from COPD patients to mice triggers increased lung inflammation in recipient mice (33). Therefore, gut microbiota and the importance of the gut-lung axis should not be neglected in future studies.

**Conclusions.** In AECOPD with acidemia, lower lung mycobiota $\alpha$-diversity is linked with the severity of the exacerbation, assessed by mortality and the requirement for invasive mechanical ventilation, whereas lung bacteriobiota $\alpha$-diversity is not. This study encourages a large multicenter cohort study investigating the role of lung microbiota in severe AECOPD and urges investigation of the role of the fungal kingdom of lung microbiota in severe AECOPD pathophysiology.

## MATERIALS AND METHODS

**Patient inclusion and data collection.** Every consecutive patient older than 18 years of age admitted for severe AECOPD to the medical ICU at the Centre Hospitalier Universitaire Bordeaux from October 2018 to March 2020 was assessed for eligibility. AECOPD was identified according to the Global Initiative for Chronic Obstructive Lung Disease Criteria (1), i.e., a change in the patients' baseline dyspnea, cough, and/ or sputum that is beyond day-to-day variations, of acute onset. Hypercapnia was defined as carbon dioxide arterial partial pressure ($PaCO_2$) $\geq$ 5.6 kPa consistent with the recent classification of severe AECOPD according to the Rome proposal (34). COPD patients admitted to the ICU for cause other than severe AECOPD, under guardianship protection, or with decisions of withholding or withdrawing intensive therapies at admission because of poor predicted prognostic were not included. Samples (induced sputum or TA) were collected within the 24 h following admission to the ICU and frozen at $-80°C$. Criteria for orotracheal intubation (OTI) were contraindicated for NIV or persistent respiratory failure (respiratory rate > 30/ min, accessory respiratory muscles activation) and/or acidemia (pH < 7.32) despite NIV.

Data were prospectively recorded by the physician(s) in charge of the patient by questioning the patients, patients' family, and patients' general practitioners. Electronic worksheets were completed by two medical intensive care residents. Comorbidities were defined as follows: COPD and asthma were defined according to prior lung function testing following GOLD guidelines (1). Chronic heart failure was defined according to prior transthoracic echocardiography and chronic coronary disease based on prior stress test or percutaneous coronary intervention. Other comorbidities included history of chronic kidney disease, immunosuppression (drugs, hematological disease, blood marrow transplantation, solid organ transplantation, plasma exchanges indicated by autoimmune disorders, human immunodeficiency virus infection), and the simplified acute physiology score II (SAPSII). Respiratory parameters were recorded at admission, and ventilation parameters were recorded as soon as the patient's condition was stabilized. Mortality was assessed at day 28 after admission to the ICU.

**DNA extraction, library preparation, and statistical analyses.** DNA extraction was performed using QIAamp PowerFaecal Pro DNA kit (Qiagen, Valencia, CA, USA). A first step of mechanical lysis (2 cycles of 30 s at 7,000 rpm on a Precellys Evolution homogenizer) was added to the chemical lysis of the kit as previously described (35). The lung microbiota and mycobiota composition of samples were assessed, respectively, by using the V3-V4 regions of the bacterial 16S rRNA encoding gene and the internal transcribed spacer 2 (ITS2) region of the fungal rRNA genes. The respective primers used to amplify these loci were as follows: 16S-forward, TACGGRAGGCAGCAG; 16S-reverse, CTACCNGGGTATCTAAT; ITS2-forward, GTGARTC ATCGAATCTTT; and ITS2-reverse, GATATGCTTAAGTTCAGCGGGT (35). Sequencing (2 × 250 bp paired-end) was performed on MiSeq sequencer (Illumina, San Diego, CA, USA) at the PGTB platform (INRAe, University of Bordeaux, Cestas, France).

The bacterial and fungal reads were demultiplexed; 16S and ITS2 primers were removed using CutAdapt, with no mismatch allowed within the primer sequences. All samples were processed through the DADA2 pipeline in R (version 4.0.3) for quality filtering and trimming, dereplication, and merging of paired-ends reads (36, 37). According to a recent evaluation (38), only forward sequences were analyzed with DADA2, and no filter other than the removal of low-quality and chimeric sequences was applied for characterizing the fungal community. We used mock communities (compositions in the Supplemental Materials) and negative controls (three from the DNA extraction step with unloaded swabs and three from the PCR amplification step) to ensure the quality of the sequencing. Two distinct amplicon sequence variants (ASV) tables were constructed, and taxonomy was assigned from the Silva database (release 138) for bacterial ASVs and the Unite database (release 8.2) for fungal ASVs. Scripts used for bioinformatics analysis are available in the Supplemental Materials. Comparison of $\beta$-diversity between negative control, mock community, and samples are available in Fig. S1 and S2 for mycobiota and bacteriobiota, respectively. The final average read counts were 19,004 (standard deviation $\pm$ 16,480) for 330 bacterial ASVs and 831 (standard deviation $\pm$ 2,406) for 58 fungal ASVs. Samples with less than 100 reads after filtration in the ITS2 analyses were not further included. The number of reads before and after filtration and rarefaction curves are provided (Table S1 and Fig. S3, respectively). The 16S rRNA gene and ITS2 sequences have been submitted to the European Nucleotide Archive (accession No. ERP134910).

For microbiota and mycobiota analysis, $\alpha$-diversity metrics (Richness, Simpson, and Shannon indices) were generated by using the phyloseq R package. For cross-sectional analyses, at a specific time, significant differences in $\alpha$-diversity were determined using the Mann-Whitney Wilcoxon rank-sum test. Between sample $\beta$-diversity differences (measured using Bray-Curtis dissimilarity) were tested using a permutational multivariate analysis of variance (PERMANOVA) from the vegan R package with 10,000 permutations, while accounting for individual identity as a covariate. Linear discriminant analysis (LDA) effect size (LefSe) analysis was performed from microbiomeMarker package. Statistical analysis was performed with the R studio program (version 1.3.1056 for Windows); correction for multiple testing was performed using the Benjamini-Hochberg false discovery rate (FDR) procedure; a *P* value or an FDR-adjusted *P* value equal to or less than 0.05 was considered statistically significant.

**Statistical analysis.** No statistical sample size calculation was performed *a priori*, and sample size was equal to the number of patients admitted to the ICU during the study period. Quantitative variables are presented as medians and interquartile range (IQRs) and compared by use of the Mann-Whitney Wilcoxon rank-sum test. Categorical variables are expressed as the number of patients (percentage) and compared by means of the chi-square or Fisher tests. All statistical tests were two-tailed, and statistical significance was defined as $P < 0.05$. Statistical analyses were assessed by the R studio program (version 1.3.1056 for Windows).

**Ethics.** According to French law and the French Data Protection Authority, the handling of these data for research purposes was declared to the Data Protection Officer of the Centre Hospitalier Universitaire Bordeaux. The study obtained the approval of the Comité de Protection des Personnes EST 1 (declaration number 2018/37). Patients (or their relatives, if any) were notified about the anonymized use of their health care data via the department's booklet. Informed consent was obtained from all patients or from their legal representatives.

**Ethics approval and consent to participate.** The study was approved by the "Comité de Protection des Personnes" EST 1 (declaration number 2018/37) and performed according to The Code of Ethics of the World Medical Association (Declaration of Helsinki).

**Data availability.** The 16S rRNA gene and ITS2 sequences have been submitted to the European Nucleotide Archive (accession No. ERP134910). The scripts used for bioinformatics analysis during the current study are available in the Supplemental Materials.

## SUPPLEMENTAL MATERIAL

Supplemental material is available online only.
**SUPPLEMENTAL FILE 1**, PDF file, 1.4 MB.

## ACKNOWLEDGMENTS

We thank Erwan Guichoux and Marie Massot for technical assistance, Frédéric Perry for assistance with clerical work, and Elizabeth Lepshina for English editing. We are grateful to every ICU health worker who cared for the patients and helped with sample collection.

We declare no conflict of interest.

This work was supported by a grant from the Fédération Girondine de Lutte contre les Maladies Respiratoires. R.P. received a personal salary grant from Centre Hospitalier Universitaire de Bordeaux (M.D./Ph.D. program).

R.P., P.B., L.D., A.B., and D.G. contributed to the conception and design of the study. P.S., A.O., P.-L.J., H.-N.B., O.P., and R.P. contributed to the acquisition of data. R.P. and A.C. performed DNA extraction. S.I. and R.E. performed bioinformatics and statistical analysis. Each author drafted or provided critical revision of the article and provided final approval of the version submitted for publication.

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
