## [Reviewer comments · Microbiology Spectrum]

Microbiology Spectrum

Lung mycobiota α -diversity is linked to severity in critically ill patients with acute exacerbation of chronic obstructive pulmonary disease

Raphaël Enaud, Pierre Sioniac, Sébastien Imbert, Pierre-Laurent Janvier, Adrian Camino, Hoang-nam Bui, Odile Pillet, Arthur Orioux, Alexandre Boyer, Patrick Berger, Didier Gruson, Laurence Delhaes, and Renaud Prével

Corresponding Author(s): Renaud Prével, CHU Bordeaux

Review Timeline:

Submission Date:	December 8, 2022
Editorial Decision:	January 23, 2023
Revision Received:	February 10, 2023
Editorial Decision:	February 28, 2023
Revision Received:	March 2, 2023
Accepted:	March 12, 2023

Editor: Soo Chan Lee

Reviewer(s): Disclosure of reviewer identity is with reference to reviewer comments included in decision letter(s). The following individuals involved in review of your submission have agreed to reveal their identity: Mariusz Dyląg (Reviewer #1); Laura Tipton (Reviewer #3)

Transaction Report:

DOI: <https://doi.org/10.1128/spectrum.05062-22>

Dr. Renaud Prével
Centre Hospitalier Universitaire de Bordeaux
bordeaux
France

Re: Spectrum05062-22 (Lung mycobiota α -diversity is linked to Day-28 mortality in critically ill patients with severe acute exacerbation of chronic obstructive pulmonary disease)

Dear Dr. Renaud Prével:

I have received the reviews of your manuscript entitled "Lung mycobiota α -diversity is linked to Day-28 mortality in critically ill patients with severe acute exacerbation of chronic obstructive pulmonary disease", and I regret to inform you that we will not be able to publish it in Spectrum. Although reviewer 1 considered this study would fit the the journal, the other reviewer raised considerable concerns about the data and the interpretation of them. By agreeing the reviewer 2's assessment, I can not be positive for your manuscript for further consideration. Your submission was read by reviewers with expertise in the area addressed in your study and it was the consensus view of these reviewers that your paper did not meet the standards necessary for publication. Copies of the reviewers' comments are attached for your consideration.

I am sorry to convey a negative decision on this occasion, but I hope that the enclosed reviews are useful. Please note, rejections from Microbiology Spectrum are final and your manuscript will not be considered by other ASM journals. We wish you well in publishing this report in another journal and hope that you will consider Spectrum in the future.

Sincerely,

Soo Chan Lee
Editor, Microbiology Spectrum

Reviewer comments:

Reviewer #1 (Comments for the Author):

Dear Authors,

in my opinion your work is very interesting in a cognitive context. This study contributes a lot to medical mycology, microbiology and pharmacology. Moreover, your work contributes a lot in the context of a better understanding of chronic obstructive pulmonary disease (COPD) and research on the lung microbiota. The Authors already have extensive experience in research topic related with human microbiome, as evidenced by their previous work (<https://doi.org/10.1186/s13054-022-03980-8>). All the tables and figures are appropriate for this type of article. In general, the paper has a logical flow and it is refined in detail. The abstract well correspond with the main aspects of the work. Nevertheless, I see one important weak point of this manuscript, namely is the conclusion (quote) "Non-survivors and patients requiring invasive mechanical ventilation have lower lung mycobiota α -diversity than survivors and patients only receiving non-invasive ventilation respectively" Can we compare non-survivors (only 4) with survivors (21 patients) to conclude that in case of the first mentioned were observed lower lung mycobiota α -diversity than in case of survivors? The Authors themselves see this problem (I quote) "The main limitation is the monocentric character of our study and the limited number of nonsurvivors". I am convinced that the Authors are able to resolve this problem and redraft the text of the manuscript accordingly, and more appropriately formulate conclusions what will be significant for this manuscript.

As a reviewer I am obligated to pay attention even to less important weak points of this work and all mentioned below comments should be carefully considered.

Line 45, page 3

To the best of my knowledge after „ventilation" and before „respectively" should be comma.

Line 56, page 3

„NIV" - all the abbreviations should be explained when used for the first time

Lines 62-64, page 4

For references should be used different type brackets (for example square brackets) to distinguish from additional information placed in parentheses. Let's check the entire manuscript in this context.

Line 140, page 7

To the best of my knowledge after „bacteriobiota" and before „respectively" should be comma.

Line 152, page 7

Delete repetition „was performed"

Line 152, page 7

As I suspect between „microbiome" and „Marker" should be space

Line 178, page 8

As I know should be „Patients' characteristics are presented in Table 1"

Lines 225-226, page 10

In my opinion, this type of writing looks better ($p=0.02$, $p=0.02$ and $p=0.03$ respectively for Richness, Shannon and Simpson indices; Figure 4 A, B, C)

Lines 269-270, page 12

In my opinion should be „The absence of observed differences ..."

Line 319, page 14

To the best of my knowledge „List of abbreviations" is unnecessary and any abbreviations used in the text of the manuscript should be explained/expanded upon first use within the text.

Figures

Figures 2-5 should be better resolution.

Reviewer #2 (Comments for the Author):

Enaud et al. investigated the microbiota of patients with lung disease COPD. Their main conclusion was that the alpha diversity of fungi was lower in non-recovered patients. The study addressed an important question. However, the methods and results need considerable improvements.

1) The main conclusion was that the non-recovered patients had lower fungal diversity. After looking into more details, I have a few concerns:

-It's unclear if the samples had enough sequencing depth. Ln 142 mentioned "samples with less than 100 reads in the ITS2 analyses were not included", which I doubt is a sufficient depth. Did the authors examine the rarefaction curves? This issue was also obvious in Fig. 2A which I believe is the most important figure of this study. Overall, the samples all had very low observed ASVs (2-7 ASVs per sample!). Without sufficient sequencing depths, this is not convincing.

-The authors provided very little taxonomy information regarding the mycobiota. The only relevant figure was Supplementary Fig. 9 which only labeled "Ascomycota" and "Basidiomycota"! Given that only 2-7 ASVs were present per sample, every ASV might play a crucial role. If possible, taxonomy assignment to the genus level can provide valuable information. Did the authors find any shared ASVs across these patients?

2) The authors provided detailed data on the patients (Table 1). However, this is completely unlinked with the microbiota. Is it possible to perform some analysis (e.g. CCA or other correlation) to link these two data? If no integration was attempted, Table one looks like a supplementary table.

3) The sample number was very different between non-recovered ($n=4$) versus recovered patients ($n=21$). I personally consider every sample precious, and sampling bias shouldn't be the reason to determine whether this is a good study. However, it does make it difficult to draw a concrete conclusion.

Overall, I encourage the authors to collaborate with a microbiologist or mycologist who is more familiar with the analyses and the interpretation of microbial taxa.

Some minor points:

-The quality of the figures needs to be improved. For example, the Y-axis of Fig. 2A was partly chopped and blurred.

-In the materials and methods: What are "16S-forward", "ITS2-forward" primers? The authors did provide the primer sequences. However, if these are common primers, the authors should use the primer name (e.g. ITS3, 505F etc.) and refer to the proper literature.

Dear Editor,

Thank you for assessing our work for potential publication in Microbiology Spectrum and for having it peer-reviewed.

We also would like to thank both Reviewers for their relevant comments which helped us to improve this manuscript.

Please find thereafter the responses we address to Reviewer's comments.

Reviewer #1 (Comments for the Author):

Dear Authors, in my opinion your work is very interesting in a cognitive context. This study contributes a lot to medical mycology, microbiology and pharmacology. Moreover, your work contributes a lot in the context of a better understanding of chronic obstructive pulmonary disease (COPD) and research on the lung microbiota. The Authors already have extensive experience in research topic related with human microbiome, as evidenced by their previous work (<https://doi.org/10.1186/s13054-022-03980-8>). All the tables and figures are appropriate for this type of article. In general, the paper has a logical flow and it is refined in detail. The abstract well correspond with the main aspects of the work. Nevertheless, I see one important weak point of this manuscript, namely is the conclusion (quote) "Non-survivors and patients requiring invasive mechanical ventilation have lower lung mycobiota α -diversity than survivors and patients only receiving non-invasive ventilation respectively" Can we compare non-survivors (only 4) with survivors (21patients) to conclude that in case of the first mentioned were observed lower lung mycobiota α -diversity than in case of survivors? The Authors themselves see this problem (I quote) "The main limitation is the monocentric character of our study and the limited number of nonsurvivors". I am convinced that the Authors are able to resolve this problem and redraft the text of the manuscript accordingly, and more appropriately formulate conclusions what will be significant for this manuscript.

As stated in the discussion part, we agree with this comment. In order to address this point, we rephrased the title, abstract, importance, highlights, discussion and conclusions sections to conclude about the severity of AECOPD -assessed by survival and requirement for invasive mechanical ventilation- and not only survival. We highlighted every change in the formulation in the manuscript.

As a reviewer I am obligated to pay attention even to less important weak points of this work and all mentioned below comments should be carefully considered.

Line 45, page 3 To the best of my knowledge after „ventilation" and before „respectively" should be comma.

Line 56, page 3 „NIV" - all the abbreviations should be explained when used for the first time

Lines 62-64, page 4 For references should be used different type brackets (for example square brackets) to distinguish from additional information placed in parentheses. Let's check the entire manuscript in this context.

Line 140, page 7 To the best of my knowledge after „bacteriobiota" and before „respectively" should be comma.

Line 152, page 7 Delete repetition „was performed"

Line 152, page 7 As I suspect between „microbiome" and „Marker" should be space

Line 178, page 8 As I know should be „Patients' characteristics are presented in Table 1"

Lines 225-226, page 10 In my opinion, this type of writing looks better ($p= 0.02$, $p= 0.02$ and $p= 0.03$ respectively for Richness, Shannon and Simpson indices; Figure 4 A, B, C)

Lines 269-270, page 12 In my opinion should be „The absence of observed differences ..."

Line 319, page 14 To the best of my knowledge „List of abbreviations" is unnecessary and any abbreviations used in the text of the manuscript should be explained/expanded upon first use within the text.

Figures Figures 2-5 should be better resolution.

We corrected every point requested by Reviewer 1, except for microbiomeMarker which is the name of the R package and for references as guide for authors indicate that this type of brackets should be used. We will be pleased to satisfy the Reviewer's request changing them to square brackets depending on Editor's preference.

Reviewer #2 (Comments for the Author):

Enaud et al. investigated the microbiota of patients with lung disease COPD. Their main conclusion was that the alpha diversity of fungi was lower in non-recovered patients. The study addressed an important question. However, the methods and results need considerate improvements.

1) The main conclusion was that the non-recovered patients had lower fungal diversity. After looking into more details, I have a few concerns:

-It's unclear if the samples had enough sequencing depth. Ln 142 mentioned "samples with less than 100 reads in the ITS2 analyses were not included", which I doubt is a

sufficient depth. Did the authors examine the rarefaction curves? This issue was also obvious in Fig. 2A which I believe is the most important figure of this study. Overall, the samples all had very low observed ASVs (2-7 ASVs per sample!). Without sufficient sequencing depths, this is not convincing.

We do agree with Reviewer's concerns comments but we think there is a misunderstanding about the sequencing depth. In fact, Reviewer 2 relevantly raised a concern about the sequencing depth. Nevertheless, the number of reads stated Ln142 regards reads after filtration in a low-biomass highly human-abundant environment and not the total number of reads after sequencing. We obtained dozen thousands of reads for each sample right after sequencing.

We apologize as we should have stated it more clearly in our Methods section and provided the number of ITS2 reads before and after decontamination. Tables presenting the number of ITS2 reads before and after filtration and the rarefaction curves are now provided in the Supplementary Materials.

-The authors provided very little taxonomy information regarding the mycobiota. The only relevant figure was Supplementary Fig. 9 which only labeled "Ascomycota" and "Basidiomycota"! Given that only 2-7 ASVs were present per sample, every ASV might play a crucial role. If possible, taxonomy assignment to the genus level can provide valuable information. Did the authors find any shared ASVs across these patients?

Our team does include two mycologists (Prof Laurence Delhaes, head of Mycology Department and Dr Sébastien Imbert). The low level of taxonomy assignment to the genus level is inherent to short-read sequencing as Unite sequence database is much less furnished than bacterial sequence database, especially for Basidiomycota for which assignment only to class is frequent.

If we run the bioinformatics analyses with agglomeration to ASVs and not genus, we obtain the same results regarding α - and β -diversity. Doing so, we can assess that ASVs are shared across a median number of 2 patients, IQR [2-4].

Looking for ASVs assigned to the specie (mostly Ascomycota), we can describe that:

Candida albicans is present in every patient and represented by 12 ASVs.

Erysiphe trifoliorum: 6 patients, 2 ASVs

Malassezia globosa: 4 patients, 2 ASVs

Cladosporium ramotenellum: 4 patients, 1 ASV

Cymatoderma caperatum: 4 patients, 1 ASV

Saccharomyces kudriavzevii: 3 patients, 1 ASV

Malassezia arunalokei : 3 patients, 2 ASVs

Malassezia restricta : 2 patients, 1 ASV

Candida dubliensis: 2 patients, 1 ASV

Xylodon detriticus : 2 patients, 1 ASV

Trametes versicolor : 2 patients, 1 ASV
Fomitopsis pinicola : 2 patients, 1 ASV
Candida bracarensis : 2 patients, 1 ASV
Kluyveromyces marxianus : 2 patients, 1 ASV

2) The authors provided detailed data on the patients (Table 1). However, this is completely unlinked with the microbiota. Is it possible to perform some analysis (e.g. CCA or other correlation) to link these two data? If no integration was attempted, Table one looks like a supplementary table.

We provided detailed data on the patients to exhibit high external validity as patients' characteristics and mortality rate are consistent with what has previously been published. Due to this reason, we think that it should not be placed as a supplementary table. Nevertheless, we do agree that, unfortunately, the relative limited size of sample does not allow multivariate regression analysis.

3) The sample number was very different between non-recovered (n =4) versus recovered patients (n = 21). I personally consider every sample precious, and sampling bias shouldn't be the reason to determine whether this is a good study. However, it does make it difficult to draw a concrete conclusion.

As stated in the discussion part, we agree with this comment which joins Reviewer 1 comment. In order to address this point, we rephrased the title, abstract, importance, highlights, discussion and conclusions sections to conclude about the severity of AECOPD -assessed by survival and requirement for invasive mechanical ventilation- and not only survival. We highlighted every change in the formulation in the manuscript. The proportion of survivors and non-survivors is consistent with what has previously been described in the literature.

Overall, I encourage the authors to collaborate with a microbiologist or mycologist who is more familiar with the analyses and the interpretation of microbial taxa. We addressed this comment in the second part of the first comment.

Some minor points:

-The quality of the figures needs to be improved. For example, the Y-axis of Fig. 2A was partly chopped and blurred.

Quality of the figures has been improved.

-In the materials and methods: What are "16S-forward", "ITS2-forward" primers? The authors did provide the primer sequences. However, if these are common primers, the authors should use the primer name (e.g. ITS3, 505F etc.) and refer to the proper literature.

These primers are the same than these standardized and optimized by Genoscreen company and we cited the appropriate reference (Vandenborgh *et al.* J Allergy Clin Immunol 2021).

We hope that these precisions will make you consider the possibility to submit a revised manuscript for potential publication in Microbiology Spectrum.

Very respectfully,

Dr Raphaël Enaud and Dr Renaud Prével

February 28, 2023

Dr. Renaud Prével
CHU Bordeaux
Place Amélie Raba Léon
Bordeaux
France

Re: Spectrum05062-22R1-A (Lung mycobiota α -diversity is linked to severity in critically ill patients with acute exacerbation of chronic obstructive pulmonary disease)

Dear Dr. Renaud Prével:

Thank you for submitting your manuscript to Microbiology Spectrum. A second set of reviewers judged your studies and found that it has a merit. However, it needs some modifications. When submitting the revised version of your paper, please provide (1) point-by-point responses to the issues raised by the reviewers as file type "Response to Reviewers," not in your cover letter, and (2) a PDF file that indicates the changes from the original submission (by highlighting or underlining the changes) as file type "Marked Up Manuscript - For Review Only". Please use this link to submit your revised manuscript - we strongly recommend that you submit your paper within the next 60 days or reach out to me. Detailed instructions on submitting your revised paper are below.

Link Not Available

Sincerely,

Soo Chan Lee

Journals Department
Reviewer comments:

Reviewer #1 (Comments for the Author):

Dear Authors,

I really appreciate your effort to update and improve your manuscript. The issues (doubts) I have raised have been carefully considered and addressed by you. First of all, thank you for rewording Discussion, Conclusions and correcting the title, which allows me to consider that the issue I reported has been resolved. Moreover, the figures included in the current version of the manuscript are really better resolution.

Thank you for your contribution to answer to my questions and concerns.
Reviewer

Reviewer #3 (Comments for the Author):

Overall Enaud, et al. have conducted and analyzed their small study of acute exacerbation of COPD well. They found that those with more severe AECOPD have lower fungal alpha diversity but no such difference among the bacterial community. Severity was defined two different ways, by survivorship and by the need for invasive mechanical ventilation, and both definitions showed the same pattern. Small sample sizes prevented a more detailed analysis, such as ASV by ASV, but this study should serve as a motivation for a larger study.

My main issues are in the formatting of the reporting:

- 1) Within the results section, breaking it up with the subheadings disrupts the flow of the manuscript and I would recommend reducing the number of subheadings by combining the alpha and beta diversity comparisons of non-survivors and survivors (ie lines 209-220), as well as those requiring mechanical ventilation and those that don't (ie lines 222-235).
- 2) The paragraph on study limitations (starting on 287) should be reworked into multiple paragraphs. I would suggest breaking it at the discussion of causality (line 299). Most clinical studies lack causality demonstration so this is not necessarily a limitation in my mind and could be reworked into a discussion of recommended future studies, which the current paragraph starts to address.
- 3) The figures need different color pallets for different comparisons. If you leave survivors and non-survivors as coral and aqua, then perhaps no_OTI should be seagreen and OTI be violetred to maintain that the more severe AECOPD is a warmer color but to also indicate that these are different divisions of the data.

Staff Comments:

Preparing Revision Guidelines

Please return the manuscript within 60 days; if you cannot complete the modification within this time period, please contact me. If you do not wish to modify the manuscript and prefer to submit it to another journal, please notify me of your decision immediately so that the manuscript may be formally withdrawn from consideration by Microbiology Spectrum.

Dear Editor,

Thank you for assessing our work for potential publication in Microbiology Spectrum and for having it peer-reviewed.

We also would like to thank both Reviewers for their relevant comments which helped us to improve this manuscript.

Please find thereafter the responses we address to Reviewer's comments.

Reviewer #1 (Comments for the Author):

Dear Authors,

I really appreciate your effort to update and improve your manuscript. The issues (doubts) I have raised have been carefully considered and addressed by you. First of all, thank you for rewording Discussion, Conclusions and correcting the title, which allows me to consider that the issue I reported has been resolved. Moreover, the figures included in the current version of the manuscript are really better resolution.

Thank you for your contribution to answer to my questions and concerns.

Reviewer

We would like to thank Reviewer 1 for his/her meaningful comments and appreciation of our manuscript.

Reviewer #3 (Comments for the Author):

Overall Enaud, et al. have conducted and analyzed their small study of acute exacerbation of COPD well. They found that those with more severe AECOPD have lower fungal alpha diversity but no such difference among the bacterial community. Severity was defined two different ways, by survivorship and by the need for invasive mechanical ventilation, and both definitions showed the same pattern. Small sample sizes prevented a more detailed analysis, such as ASV by ASV, but this study should serve as a motivation for a larger study.

My main issues are in the formatting of the reporting:

1) Within the results section, breaking it up with the subheadings disrupts the flow of the manuscript and I would recommend reducing the number of subheadings by combining the alpha and beta diversity comparisons of non-survivors and survivors (ie lines 209-220), as well as those requiring mechanical ventilation and those that don't (ie lines 222-235).

2) The paragraph on study limitations (starting on 287) should be reworked into multiple paragraphs. I would suggest breaking it at the discussion of causality (line 299). Most clinical studies lack causality demonstration so this is not necessarily a limitation in my mind and could be reworked into a discussion of recommended future studies, which the current paragraph starts to address.

We have addressed both of Reviewer's comments as highlighted in our manuscript combining the alpha and beta diversity comparisons of non-survivors and survivors and of those requiring mechanical ventilation and those who don't and re-organizing the discussion section as suggested by Reviewer 3.

3) The figures need different color pallets for different comparisons. If you leave survivors and non-survivors as coral and aqua, then perhaps no_OTI should be seagreen and OTI be violetred to maintain that the more severe AECOPD is a warmer color but to also indicate that these are different divisions of the data.

We changed the color pallet as suggested by Reviewer 3 for Figures 4 and 5.

We hope that these precisions will make you consider the possibility to submit a revised manuscript for potential publication in Microbiology Spectrum.

Very respectfully,

Dr Raphaël Enaud and Dr Renaud Prével

March 12, 2023

Dr. Renaud Prével
CHU Bordeaux
Place Amélie Raba Léon
Bordeaux
France

Re: Spectrum05062-22R2 (Lung mycobiota α -diversity is linked to severity in critically ill patients with acute exacerbation of chronic obstructive pulmonary disease)

Dear Dr. Renaud Prével:

Your manuscript has been accepted, and I am forwarding it to the ASM Journals Department for publication. You will be notified when your proofs are ready to be viewed.

Sincerely,

Soo Chan Lee
Editor, Microbiology Spectrum
